# Transcriptional Inflammatory Signature in Healthy Donors and Different Radiotherapy Cancer Patients

**DOI:** 10.3390/ijms25021080

**Published:** 2024-01-16

**Authors:** Gráinne O’Brien, Malgorzata Kamuda, Lourdes Cruz-Garcia, Mariia Polozova, Ales Tichy, Marketa Markova, Igor Sirak, Oldrich Zahradnicek, Piotr Widłak, Lucyna Ponge, Joanna Polanska, Christophe Badie

**Affiliations:** 1Cancer Mechanisms and Biomarkers Group, Centre for Radiation, Chemical and Environmental Hazards, UK Health Security Agency, Oxfordshire OX11 0RQ, UK; grainne.obrien@ukhsa.gov.uk (G.O.); lourdes.cruzgarcia@ukhsa.gov.uk (L.C.-G.); maria.polozova@ukhsa.gov.uk (M.P.); 2Department of Data Mining, Silesian University of Technology, 44-100 Gliwice, Polandjoanna.polanska@polsl.pl (J.P.); 3Department of Radiobiology, Faculty of Military Health Sciences in Hradec Králové, University of Defence, 662 10 Brno, Czech Republic; 4Biomedical Research Centre, University Hospital Hradec Králové, 500 05 Hradec Králové, Czech Republic; 5Institute of Hematology and Blood Transfusion, 128 00 Praha, Czech Republic; marketa.markova@uhkt.cz; 6Department of Oncology and Radiotherapy and 4th Department of Internal Medicine—Hematology, University Hospital, 500 05 Hradec Králové, Czech Republic; igor.sirak@fnhk.cz; 7Department of Radiation Dosimetry, Nuclear Physics Institute, Czech Academy of Sciences, 180 00 Prague, Czech Republic; zahradnicek@ujf.cas.cz; 8Clinical Research Support Centre, Medical University of Gdańsk, Gdańsk, M. Skłodowskiej-Curie 3a Street, 80-210 Gdańsk, Poland; piotr.widlak@io.gliwice.pl; 9Maria Skłodowska-Curie National Research Institute of Oncology, 44-102 Gliwice, Poland; lucyna.ponge@io.gliwice.pl

**Keywords:** inflammation, gene expression, biomarkers, human blood, cancer patients, radiotherapy

## Abstract

Cancer and ionizing radiation exposure are associated with inflammation. To identify a set of radiation-specific signatures of inflammation-associated genes in the blood of partially exposed radiotherapy patients, differential expression of 249 inflammatory genes was analyzed in blood samples from cancer patients and healthy individuals. The gene expression analysis on a cohort of 63 cancer patients (endometrial, head and neck, and prostate cancer) before and during radiotherapy (24 h, 48 h, ~1 week, ~4–8 weeks, and 1 month after the last fraction) identified 31 genes and 15 up- and 16 down-regulated genes. Transcription variability under normal conditions was determined using blood drawn on three separate occasions from four healthy donors. No difference in inflammatory expression between healthy donors and cancer patients could be detected prior to radiotherapy. Remarkably, repeated sampling of healthy donors revealed an individual endogenous inflammatory signature. Next, the potential confounding effect of concomitant inflammation was studied in the blood of seven healthy donors taken before and 24 h after a flu vaccine or ex vivo LPS (lipopolysaccharide) treatment; flu vaccination was not detected at the transcriptional level and LPS did not have any effect on the radiation-induced signature identified. Finally, we identified a radiation-specific signature of 31 genes in the blood of radiotherapy patients that were common for all cancers, regardless of the immune status of patients. Confirmation via MQRT-PCR was obtained for BCL6, MYD88, MYC, IL7, CCR4 and CCR7. This study offers the foundation for future research on biomarkers of radiation exposure, radiation sensitivity, and radiation toxicity for personalized radiotherapy treatment.

## 1. Introduction

The transcriptional response to ionising radiation (IR) exposure has been widely investigated, mainly in blood, and the gene expression assay has become an established and sensitive technique for identifying individuals exposed to IR. Investigation into the transcriptional response to ionising radiation has largely focused on the DNA damage response and downstream pathways activated by the transcription factor p53 [1,2]. The inflammation process is crucial for radiation response. RT-elicited inflammation is a decisive factor in a patient’s response to therapy and has both immunosuppressive and immunostimulatory effects [3]. It depends on the radiation therapy (RT) dose, the number of fractions, and the factors which affect immune status (gender, age, accompanying diseases, individual radiosensitivity of normal tissues, and radioresistance of tumour) [4].

Until recently, studies on the identification of inflammatory biomarkers of radiation exposure have been limited and focused on in vitro transcriptional changes [5,6]. A pilot study of in vivo irradiated blood from 20 cancer patients with a range of cancer types has previously identified inflammatory genes of interest [7]. Another study on 23 cancer patients identified eight immune and inflammation-related plasma proteins whose secretion levels were altered before and after radiotherapy [8]. Results from small cohorts are linked to greater statistical uncertainty, which could lead to false negatives, consequently missing important statistically significant differences. Patients’ samples are very valuable in research but are also extremely difficult to obtain; therefore, it is not always possible to recruit the desired number of patients in such studies, leading to a weaker statistical power of the results. In this study, we gathered existing and newly generated data, including data from 63 radiotherapy patients to overcome this issue. We focused on investigating the inflammatory response in a larger cohort of healthy donors and cancer patients (Figure 1) by building on a study we previously performed [9] and on the whole blood from 10 endometrial and 8 head and neck cancer patients to obtain an inflammation-associated gene expression signature specific to ionizing radiation exposure and independent of cancer type. Moreover, we tested the reproducibility of the signature by monitoring the same healthy donors over three consecutive weeks and addressed potential confounding effects to demonstrate the specificity of this signature. 

## 2. Results

Transcriptional analysis of inflammatory genes showed variability in blood taken from healthy donors at three different intervals. This intra-variability, however, was less than the inter-variability between donors, as illustrated by the histogram clustering in Figure 2A. The unsupervised clustering showed that each individual is identifiable via an inflammatory signature.

Then, we addressed the potentially confounding effect of concomitant infection in vivo by analysing blood samples from healthy donors before and after flu vaccination or in vitro before and after exposure to LPS on the transcriptional expression of the inflammation genes analysed. Signal log ratio analysis identified 27 genes (CCL2, CXCL2, CXCL3, IL6, MAFF, IL1A, IL12B, CSF3, CXCL9, CCL3, C3, CCL19, CCL20, IL1B, C3AR1, RIPK2, PTGS2, TNF, IRF1, CD40, IFNG, HLA-DRA, OAS2, CCL4, CCL22, CFB, and CXCL10) that were significantly up-regulated after exposure to LPS in comparison to normal expression, while no genes were significantly differentially expressed after flu jabs (Figure 2B). Importantly, a UMAP-based visualisation of the healthy donor samples in comparison to cancer patient samples taken before radiotherapy was performed. This analysis reported no difference in inflammatory expression between the two groups (Figure 2C). 

Bioinformatic analysis of cancer patients’ nCounter data was performed by combining previous historical data of endometrial and head and neck cancer patients with additional head and neck and IMRT prostate cancer patient samples, and a new bioinformatic approach was applied. Different treatment times were combined into two groups: preexposure and 24 h after the last fraction (~5 weeks). A total of 31 genes (Table 1) common across all cancer types (15 up-regulated and 16 down-regulated genes) were significantly modified.

Confirmation via MQRT-PCR was obtained on a limited number of several up- and downregulated genes (Figure 3) as well as for ARG1 and BCL2L1 genes, which were identified in a previous study [9]. Significance in gene expression was calculated in comparison to preexposure samples. BCL6 (Figure 3A), MYD88 (Figure 3B), ARG1 (Appendix A), and BCL2L1 (Appendix A) were significantly up-regulated in comparison to preexposure samples; significance was mainly observed at the last timepoint during treatment (at ~4–8 weeks after the 1st fraction) or 1 month after the end of treatment. MYC (Figure 3C), IL7 (Figure 3D), CCR4 (Figure 3E), CCR7 (Figure 3F), CD40 (Appendix A), ELK1 (Appendix A), and PRKCA (Appendix A) were significantly downregulated in comparison to preexposure samples. CD40 and ELK1 only had significant early expression after 24 h and 48 h while CCR4, CCR7, IL7, and PRKCA had significant expression at later time points: ~4–8 weeks after the 1st fraction or 1 month after the end of treatment. 

Significance in gene expression was also calculated in the prostate subgroups, IMRT and CK, with a large difference observed in dose (IMRT 2 Gy v CyberKnife 7.25 Gy) and dose fractions (IMRT 39 fractions v CyberKnife 5 fractions) between groups. A significant difference was observed after IMRT treatment for genes BCL6 (Figure 4A), IL7 (Figure 4D), CCR4 (Figure 4E), CCR7 (Figure 4F), ARG1 (Appendix A), and BCL2L1 (Appendix A), while there was no significant difference in gene expression after CK treatment for these genes. In contrast, only MYD88 (Figure 4B) and CD40 (Appendix A) had a significant up- and down-regulation, respectively, in CyberKnife patient samples and no significant change in expression in IMRT patient samples. ELK1 had no change in expression in both groups (Appendix A). Both patient groups had significant downregulation in expression in CCR7 (Figure 4F), MYC (Figure 4C), and PRKCA (Appendix A) during and after treatment.

## 3. Discussion

In the context of radiotherapy treatment, we identified 15 up-regulated and 16 downregulated commonly altered genes, regardless of cancer type, some of which were confirmed via MQRT-PCR (*BCL6*, *MYD88*, *CCR4*, *CCR7*, *CD40*, *ELK1*, *IL7*, *MYC*, and *PRKCA*). These genes are specific to radiation and were not activated in response to LPS or after a flu vaccination.

BCL6, a multifunctional member of the BTB-zinc finger protein family, regulates the cell cycle and apoptosis, development, proliferation, and differentiation of B- and T-lymphocytes and inflammatory signals in macrophages [10,11]. Li et al. found that depletion of BCL6 in Tregs enhances antitumor response and delays tumour progression [12]. This gene was previously identified as radiation-responsive in our previous study in blood from radiotherapy patients at the end of their treatment, after receiving numerous fractions reaching cumulative doses of 37.38–57 Gy [7]. A similar response has been observed in this extended study for all the different cancer types. 

As exhibited in different cancer model systems, MYD88 directs innate immune signalling through the TLR members (except TLR3) and the IL-1 family and can function doubly in pro- and anti-tumorigenic responses [13]. The Toll-like receptor (TLR) family of genes, including TLR1, TLR4, TLR5, and TLR8, plays an important role in the human response to radiation exposure. Specifically, TLR1 and TLR8 are up-regulated in response to radiation exposure, leading to the production of proinflammatory cytokines, while TLR4 and TLR5 may play radioprotective roles [14,15,16]. 

Two genes, *CD40* and *ELK1*, had significant downregulation at the early time points of 24 h and 48 h. CD40 belongs to the TNF receptor family. It is expressed mostly on APCs, especially on dendritic cells and is essential for their proliferation and activation. The ligand for CD40 is CD40 ligand (CD40LG), which is expressed on CD4 T cells, CD8 T cells, B cells, and NK cells; therefore, CD40 mediates the antigen-specific activation of naive lymphocytes. The CD40/CD40L axis produces the upregulation of co-stimulatory molecules and the release of proinflammatory cytokines that leads to enhanced antitumor activity [17]. CD40LG was also identified to be downregulated by radiation in our preliminary studies [7]. The downregulation was observed at the end of the radiotherapy treatment, after 37.38–57 Gy cumulative doses. ELK1 is a known regulator of the expression of transcription factors engaged in cell growth, migration, differentiation, and survival. ERK/ELK1 signalling pathway is involved in immune cell cycle progression, while T- and B-lymphocytes and natural killer and plasma cells, principal players in anticancer immunity, strongly require ELK1 for its differentiation programs [18,19,20,21]. The significant downregulation of *CD40* and *ELK1* after 24 h and 48 h identifies two potential genes of interest as early biomarkers of treatment progress.

Interestingly, most genes (*CCR4*, *CCR7*, *IL7*, and *PRKCA*) had significant downregulation at the later time point of 4–8 weeks. CCR4 and CCR7 are the G protein-coupled receptor family’s coding genes that recognise CC chemokines. CCR4 is expressed on different immune cells, especially T-helper type 2 cells and T-regs [22]. Tumour cells, as well as tumour-associated macrophages and dendritic cells, produce high amounts of CCR4 ligands (CCL17 and CCL22) in breast, ovarian, and lung cancer patients [23,24]. High CCR4 expression in the tumour or its microenvironment is a poor prognostic indicator in lung adenocarcinoma, renal cell carcinoma, gastric, breast, and oral tongue cancer [25,26,27,28,29]. Meanwhile, other studies indicated that head and neck cancer patients with high CCR4 expression had a better prognosis [30,31]. Zhong et al. suggested that some of the chemokines, including CCR4, may be potential therapeutic targets for radiation-induced lung toxicity [32]. It has been suggested that CCR7 plays binary roles in cancer; the overexpression of the CCR7/CCL21 axis is associated with lymph node metastasis of various cancer types, and at the same time, CCR7 tends to potentiate immune cell movement to tumours [33,34]. 

Biomarkers of immune response to radiation have been explored, including circulating cytokines. Christensen et al. showed that changes in specific cytokines’ levels over baseline were associated with increased gastrointestinal and genitourinary toxicity in patients undergoing intensity-modulated radiotherapy (IMRT) for prostate cancer [35]. IL7 is a cytokine involved in T- and B-cell development. Some studies have suggested that an inadequate supply of IL-7 in the secondary lymphoid organs might be insufficient to support the survival of activated T-cells, thereby aggravating cancer immunosuppression [36,37]. Also, it is necessary to expand our understanding of the grade of neoplastic progression resulting in differing cytokine expression.

*MYC* is the only gene that demonstrated significant downregulation at every time point studied. MYC is required for the activation and cell cycle initiation of T- and B-lymphocytes and participates in several facets of the inflammatory process as well [38,39]. Activated T-lymphocytes have a constant high rate of MYC degradation; hence, when T-cell protein synthesis and amino acid uptake are restricted, MYC protein expression declines [40]. This process may happen under radiation-provoked cellular stress–DNA damage. Given that protein synthesis is one of the most energy-consuming processes in the cell, the transitory inhibition of general protein synthesis is a cellular response to stress [41]. MYC has also shown clear responsiveness to radiation in our previous studies in different cancer-type patients [7,9].

PRKCA is predominantly expressed in T-cells, determining the magnitude of the T-cell proliferative response upon T-cell activation. It has a broad spectrum of functions in many tissues, such as physiological cell processes, tumorigenesis, and inflammation. It appears to be the significant PRKC isoform involved in regulating IL-2 receptor expression [42]. A specific role of PRKCA in radiotherapy-elicited toxicities was highlighted by the study by Weigel et al., where protein and mRNA expression data indicated increased expression of PRKCA in fibroblasts of breast cancer patients developing radiation fibrosis [43]. 

Lastly, we showed that prostate cancer patients treated with IMRT had significant changes in the expression of more inflammatory genes in comparison to patients treated with CyberKnife. This could be because CyberKnife, which uses an approach called stereotactic body radiation therapy (SBRT), allows for more precise targeting of tumours while minimising damage to surrounding tissue, while there is a larger treatment site and increased dose fractions associated with IMRT. A retrospective study by Yu J. et al. found that SBRT was associated with slightly higher incidences of gastrointestinal toxicity 6, 12, and 24 months after post-radiotherapy initiation, in comparison to IMRT [44]. Another study by Pan et al., however, found that the toxicity profiles of SBRT and IMRT were similar [45]. Further research is needed to fully evaluate the long-term effects of both treatments.

Inflammation can be triggered by many diseases, including bacterial or viral infections, autoimmune conditions, and cancers, leading to a “noisy” background transcriptional expression of inflammation-related genes. As this could affect the specific response to IR during radiotherapy treatment, we addressed this potential question by studying intra-individual fluctuations of expression; remarkably, unsupervised clustering of the four healthy donors studied suggests that each individual is different enough to be identifiable via an inflammatory signature, in a presumable “inflammation-free” context. We also showed that flu jabs do not trigger an inflammation strong enough to be detected, at least with the tools we used. On the contrary, an LPS exposure ex vivo, mimicking septicaemia, can trigger an inflammation reaction detected in this study, activating 27 genes. Although one of the 27 genes, CD40, was common in gene signatures of the LPS and radiotherapy patient samples, it showed opposite responses, which were an upregulation during LPS treatment and downregulation after radiation exposure. Therefore, the confounding effects tested in this study demonstrate that we have identified a radiation-specific signature of 31 genes. 

Our study bears some limitations. The health donor control samples used for comparison to cancer patients were not similar in age due to the difficulties in obtaining healthy elderly male blood donors. Care was taken to ensure the healthy controls were recruited from similar geographic populations with similar male-to-female ratios. Also, there were no toxicity data for these patient samples to analyse in relation to gene expression. To obtain a sufficient number of samples for each toxicity grade (grades 1–5), a vastly larger study with long-term banking of samples and follow-up for toxicity symptoms would be required. Linking these data to the toxicity in radiotherapy patients would be extremely valuable, but this requires a large-scale study to validate these genes.

In summary, we have identified a panel of 31 inflammation-associated genes in radiotherapy patients. Although their usefulness remains to be confirmed, their potential to inform in ‘real-time’ for treatment monitoring, normal tissue toxicity, and morbidity prevention is promising.

## 4. Materials and Methods

### 4.1. Bioethics

The collection of blood samples from healthy donors was carried out with informed consent in accordance with the ethical approval of the West Midlands-Solihull Research Ethics Committee (REC 14/WM/1182) at CRCE, Oxfordshire. The collection of blood samples from endometrial and head and neck cancer patients was performed at the University Hospital in Hradec Kralove (Czech Republic). This study was carried out in accordance with the recommendations of The Code of Ethics of the World Medical Association-Declaration of Helsinki (approval no: 201401-S15P) with written informed consent from all subjects. The protocol was approved by the Ethical Committee of the University Hospital in Hradec Kralove (Czech Republic). The collection of blood samples from prostate and head and neck cancer patients was carried out in accordance with the Bioethical Committee in Maria Sklodowska-Curie Institute, Warszaw, with approval number 27/2015 obtained on 18 August 2015. All subjects provided written informed consent in accordance with the Declaration of Helsinki.

### 4.2. Blood Collection—Healthy Donors

Blood samples were collected from three groups of healthy individuals for different purposes: 4 healthy donors (3 females and 1 male, aged 25–54, with codes H20, H61, H64, and H65) for variability studies, 7 healthy donors (5 females and 1 male, 35 to >54 years old) for studies on confounding effects by infection by LPS and flu vaccine exposure, and 12 normal healthy donors (5 female, 7 male, aged 25–62) as control samples for cancer patient data. For variability, LPS, and flu vaccine studies, blood was collected into EDTA tubes from UKHSA, CRCE, Oxfordshire. For variability studies, blood was collected on 3 separate occasions, each 3 weeks apart, and the blood was mixed with 1 mL of RNAlater following an incubation time of 24 h at 37 °C and stored at −80 °C until further processing. 

Blood samples from 12 normal healthy individuals, used as controls for cancer patient samples, were also collected in PAXGene RNA blood collection tubes from the University Hospital in Hradec Kralove (Czech Republic).

### 4.3. Blood Collection—Cancer Patients

Blood samples from cancer patients were taken before treatment and at different time points during treatment into PAXGene blood collection tubes. Details of radiotherapy dose and patient details are provided in Table 2. The radiotherapy treatment timeline and blood collection points are collected in Figure 5. This study involved a total of 63 cancer patients with 3 different cancer types, namely, endometrial cancer, head and neck cancer, and prostate cancer. Blood samples from 10 endometrial cancer patients (aged 57–79) and 8 head and neck cancer patients (1 female and 7 males, aged 51–81) were collected from the University Hospital in Hradec Kralove (Czech Republic). Blood samples from 23 head and neck cancer patients were collected from the Maria Sklodowska-Curie Institute–Oncology Center in Poland (14 men and 9 women, aged 43–79). Blood samples from 22 prostate cancer patients were collected from the Maria Sklodowska-Curie Institute–Oncology Center in Poland. Eleven prostate cancer (aged 62–83) patients were treated with photon intensity-modulated radiation therapy (IMRT) (volumetric arc therapy—VMAT, energy: 6 MV, and dose rate: 3 Gy/min) with a daily fraction of 2 Gy at a frequency of five times a week, according to the conventional irradiation scheme. Blood samples from a further 11 prostate cancer patients (aged 52–75) were collected from the Maria Sklodowska-Curie Institute–Oncology Center in Poland. These eleven prostate cancer patients were treated with SABR using a CyberKnife (CK) (Accuray Inc., Chesapeake Terrace Sunnyvale, CA, USA) treatment unit according to the scheme 5 fractions of 7.25 Gy every second day (energy of 6 MV and dose rate of 9 Gy/min). Due to the different schemes, these CK prostate cancer patients were not included in the bioinformatic combined analysis for gene identification.

### 4.4. Blood Stimulated with LPS or Flu Vaccine

Peripheral blood from 7 healthy donors (5 females and 1 male, 35 to >54 years old) was collected in EDTA-coated tubes and incubated with LPS (500 ng/mL). LPS was prepared in 50% ethanol (stock solution 1 mg/mL) and added to 500 µL of blood. The blood samples were kept at 37 °C in an incubator with 5% CO_2_ for 24 h after exposure to allow cells to undergo DNA repair. After the incubation time, the blood was mixed with 1 mL of RNAlater and stored at −80 °C until being processed for RNA extraction. Peripheral blood samples from the same 7 healthy donors and those with LPS exposure, were collected before and approximately 24 h after they received a flu vaccine.

### 4.5. RNA Extraction

RNA was extracted from the healthy donor blood variability samples, flu vaccine samples, and LPS exposure samples preserved with RNALater using the Ribopure RNA extraction kit according to the manufacturer’s instructions. RNA was extracted from the PAXGene tube samples from cancer patients and the PAXGene tube healthy donor controls using the PAXGene Blood miRNA Kit (Qiagen, PreAnalytiX GmbH, Hilden, Germany) according to the manufacturer’s protocol. 

### 4.6. nCounter Analysis

Samples were analysed via the nCounter Analysis System (NanoString Technologies^®^, Inc., Seattle, WA, USA) according to the guidelines. The samples were run using 100 ng RNA per sample on the Human Inflammation V2 panel consisting of 249 inflammatory genes.

### 4.7. Data Analysis

To study the variability of inflammatory gene expression data in healthy donors, i.e., H20, H61, H64, and H65, unsupervised agglomerative hierarchical clustering with complete linkage and the Spearman rank correlation as the distance measure was performed on nCounter z-score normalised data from the triplicate blood samples collected at intervals of 3 weeks apart.

The Gaussian Mixture Model (GMM) classification method was used to determine the impact of ex vivo LPS treatment and flu vaccination on the response of the panel of inflammatory genes of interest and to examine the similarity of the responses. The relative nCounter gene expression values, defined as the signal log ratio (SLR), were calculated for every gene, separately for the healthy donor flu vaccine (log_2_(Flu-vaccinated/Control_Flu-vaccinated_)) and ex vivo LPS treated samples (log_2_(LPS/Control_LPS_)). Then, to compare the results between flu-vaccinated and LPS conditions, the cross-condition relative SLRs defined as log_2_((LPS/Control_LPS_) /(Flu-vaccinated/Control_Flu-vaccinated_)) was calculated for each gene. An empirical probability density function of the cross-condition relative SLR was then represented by the mixture of normal distributions (GMM). The optimal number of components was selected based on the Bayesian Information Criterion. The mixture component with the expected value close to zero represented the subset (cluster) of genes with similar responses independent of the experimental condition (flu-vaccinated and LPS-treated), while the remaining components modelled the up- and down-regulated gene subsets. The maximum posterior conditional probability criterion was used to classify genes as up-regulated, having a similar response, and down-regulated. 

Uniform Manifold Approximation and Projection for Dimension Reduction (UMAP) technique allowed for dimensionality reduction and spatial visualisation of all samples. UMAP projection was also used to investigate the heterogeneity of gene expression patterns. The preexposure samples of each cancer type (10 endometrium, 31 head and neck, 22 prostate) and 12 control samples from healthy donors were used. The batch effect was corrected using the ComBat algorithm. The corrected data were then subjected to Principal Component Analysis (PCA), and 54 PCA components, explaining 99% of the variance within the data set, were chosen for the next step. Then, the nonlinear UMAP-based dimensionality reduction algorithm was applied, providing a final two-dimensional data representation.

For combined analysis of the nCounter data, only the samples measured at two common time points (preexposure and 24 h after the last fraction (~5 weeks)) were used. Historical data of 10 endometrial and 8 head and neck cancer patients were integrated with 23 head and neck and 11 IMRT prostate cancer patient samples. Due to the specific protocol used in the treatment of CyberKnife (high doses and shorter therapy time of ~1 week), prostate cancer patients treated via CyberKnife were excluded from the overall analysis.

A paired *t*-test was performed on 249 genes separately for samples from each treatment protocol. Together with the *p*-values, Cohen’s d effect size measures were calculated and then qualified as the effect of at least medium or low impact [46]. Significantly up- and down-regulated genes were integrated across all studied cancer types.

### 4.8. Gene Expression Analysis

Reverse transcriptase reactions were performed using a High Capacity cDNA Reverse transcription kit, (Applied Biosystems, FosterCity, CA, USA) according to the manufacturer’s protocol with 350 ng of total RNA. MQRT-PCR was performed using Rotor-Gene Q (Qiagen, Hilden, Germany) on 10 endometrial cancer patients, 31 head and neck cancer patients, and 22 prostate cancer patients at all time points. All reactions were run in triplicate using PerfeCTa^®^ MultiPlex qPCR SuperMix (Quanta Biosciences, Inc. Gaithersburg, MD, USA) with primer and probe sets for target genes at 300 nM concentration each. 3′6-Carboxyfuorescein (FAM) and Texas Red (Eurogentec Ltd., Fawley, Hampshire, UK) were used as fluorochrome reporters for the hydrolysis probes analysed in multiplexed reactions with 2 genes per run, including the control. The cycling parameters were 2 min at 95 °C, followed by 45 cycles of 10 s at 95 °C and 60 s at 60 °C. Data were collected and analyzed using the Rotor-Gene Q Series Software. Gene target Ct (cycle threshold) values were normalized to a Hypoxanthine-Guanine phosphoribosyltransferase 1 (*HPRT1*) internal control. Ct values were converted to transcript quantity using standard curves obtained by serial dilution of PCR-amplified DNA fragments of each gene. The linear dynamic range of the standard curves covering six orders of magnitude (serial dilution from 3.2 × 10^−4^ to 8.2 × 10^−10^) provided PCR efficiencies between 93% and 103% for each gene with R^2^ > 0.998. Relative gene expression levels after irradiation were determined. A limited number of genes were selected for validation. Primer designs are listed in Appendix A. Statistical analyses for MQRT-PCR were performed as previously described [47]. Descriptive statistics (mean value and standard deviation, with 95% confidence intervals) were calculated in Minitab 17. Gene expression data were log2 transformed. Shapiro-Wilk test was applied to check distribution normality, while the homogeneity of variances across groups was analysed using the Bartlett’s test. Depending on the parametric tests’ assumption violation, the hypothesis on equality of population mean values was verified by the parametric *t*-test or ANOVA with Tukey HSD (Honestly Significant Difference) post hoc tests for normally distributed homogenous features and nonparametric the Mann–Whitney test or Kruskal–Wallis ANOVA with Conover post hoc test for each time point independently. No correction for multiple testing was applied. Results with *p*-value < 0.05 were treated as significant.

## Figures and Tables

**Figure 1 ijms-25-01080-f001:**
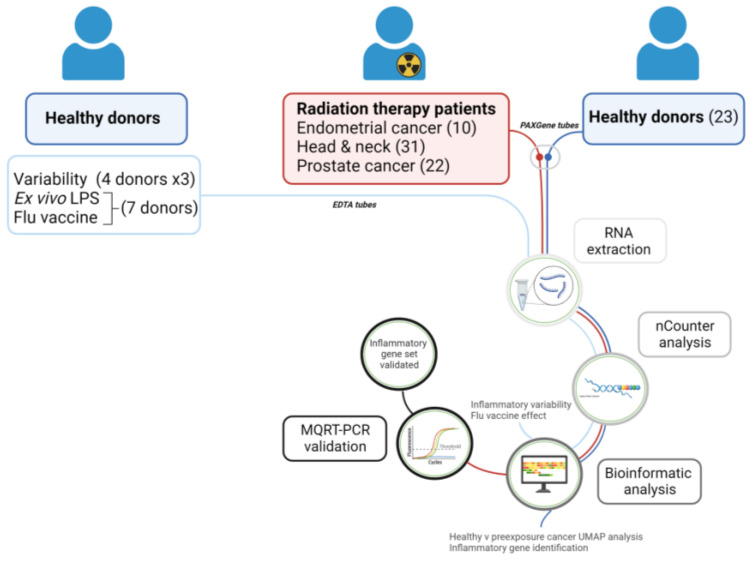
The study design of the 63 cancer patients and 23 healthy donors involved in this study and the experimental pathway of nCounter analysis, bioinformatic analysis, and MQRT-PCR validation.

**Figure 2 ijms-25-01080-f002:**
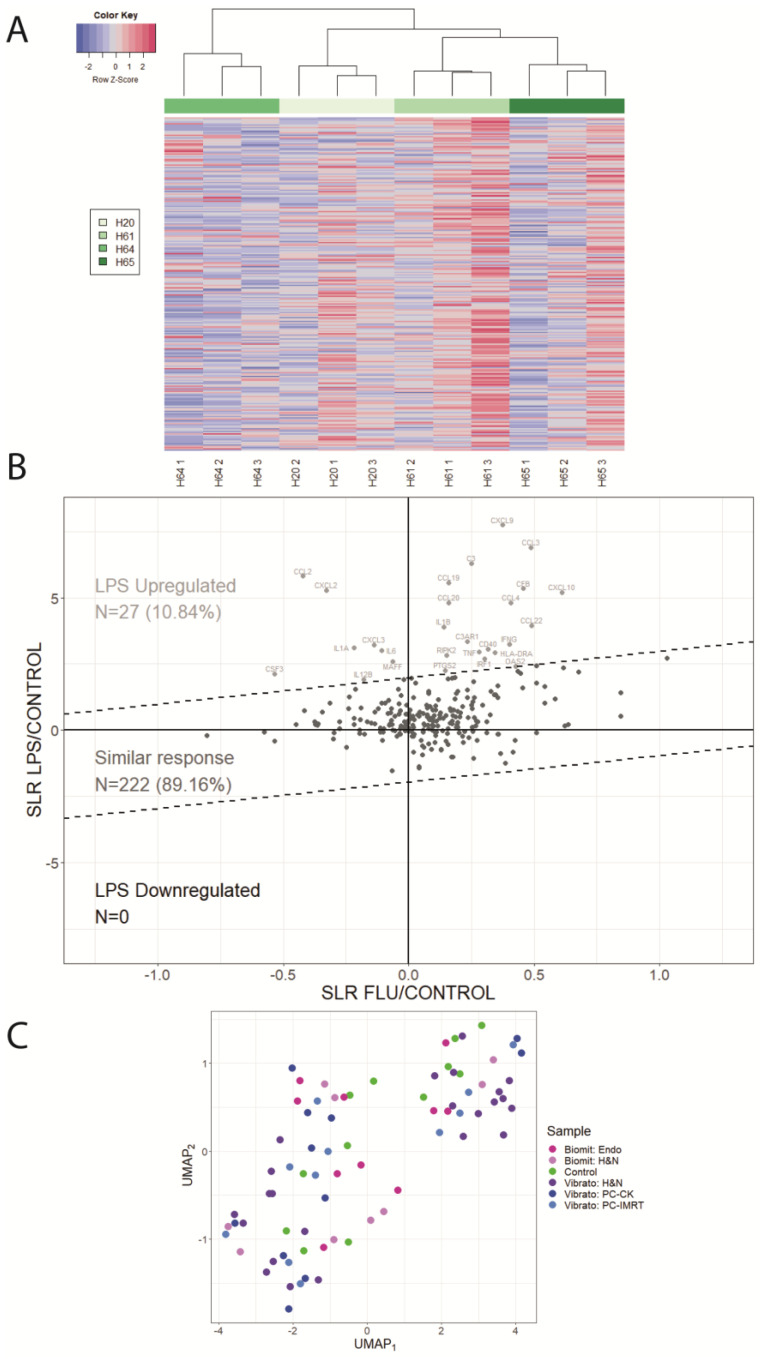
Bioinformatical analysis of health donors’ gene expression profiles. (**A**) Normal variability of inflammatory genes. nCounter expression analysis of human inflammation genes in 4 healthy human blood donors (H20, H61, H64, and H65) repeated 3 times at 3-week intervals. Unsupervised clustering and Spearman correlation as the distance method were performed on the z-score normalized data, and a complete hierarchical clustering method was used. The negative row z-score (below −2) is shown in dark blue, and the positive row z-score (above +2) is shown in dark red. (**B**) Scatterplot showing a comparison of expression changes in the blood from 7 healthy donors treated ex vivo with LPS (y-axis) and in the blood from 7 healthy donors who received a flu vaccine (x-axis). The thresholds obtained in the analysis of the Gaussian mixture model are shown in the figure with a dashed line. Up-regulated genes are shown in light grey, with similar responses shown in dark grey. N represents the number of genes in each category, and the percentage of all genes in a given group is provided in parentheses. The uniform dispersion of points suggests no significant difference between expressions from different samples. (**C**) UMAP analysis of nCounter inflammatory expression in human healthy donor controls vs cancer patient samples taken before radiotherapy. The preexposure samples of each cancer type (10 endometrium, 31 head and neck, and 22 prostate) and control samples from 12 healthy donors were used. For the collected nCounter data, a batch effect correction was performed. The first 54 PCA components, which explained about 99% of the observed variance, were selected for UMAP-based dimensionality reduction. Each point represents a different sample.

**Figure 3 ijms-25-01080-f003:**
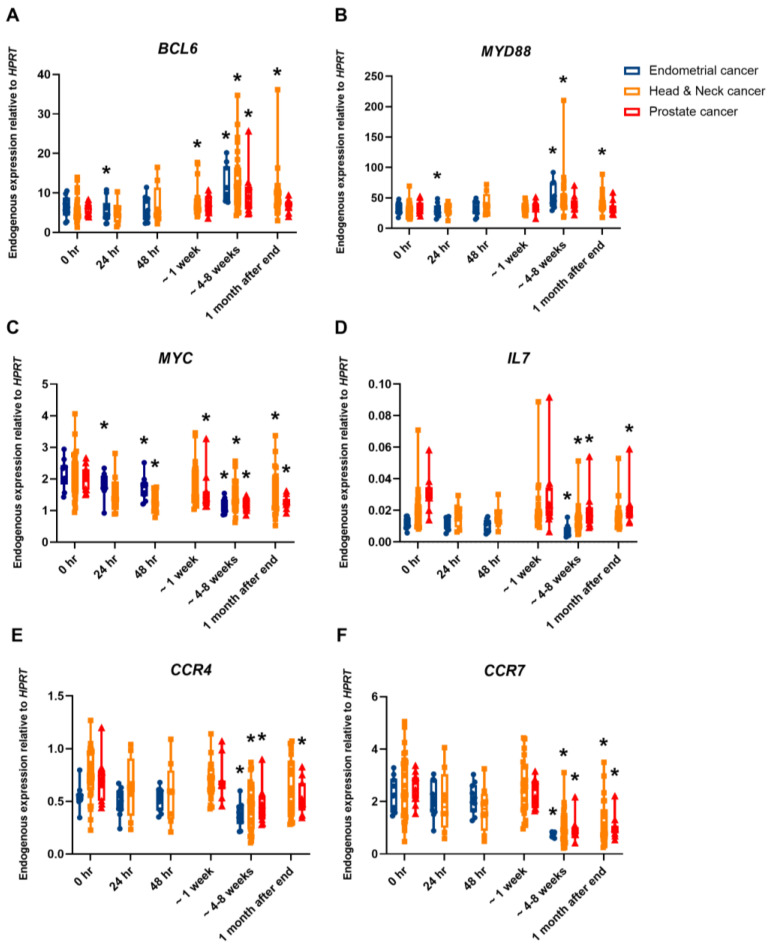
MQRT-PCR expression of the genes (**A**) BCL6, (**B**) MYD88, (**C**) MYC, (**D**) IL7, (**E**) CCR4, and (**F**) CCR7 in the whole blood of endometrial cancer patients (blue), head and neck cancer patients (orange), and prostate cancer patients (red) preexposure, 24 h after the 1st fraction and 48 h after the 1st fraction, 5th/6th fraction (~1 week), and last fraction (4–8 weeks) and 1 month after the end of the treatment. The box plot shows expression analysed in 10 endometrial cancer patients, 31 head and neck cancer patients, and 11 prostate cancer patients treated with IMRT. Expression levels are normalised to HPRT. Statistical analysis was performed on log-transformed data. * Significantly different (*t*-test, *p* < 0.05).

**Figure 4 ijms-25-01080-f004:**
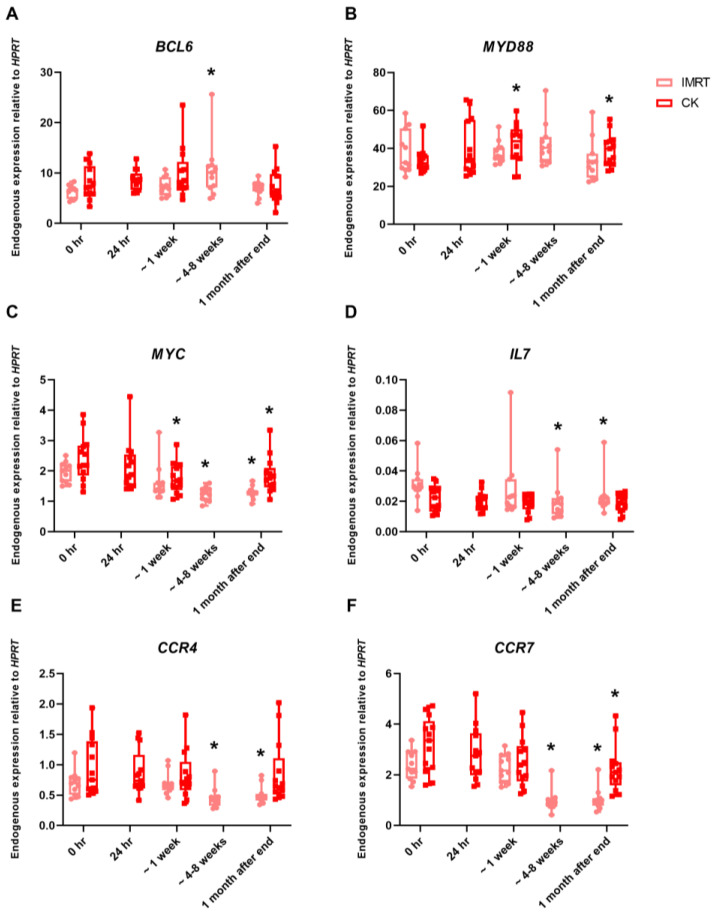
MQRT-PCR expression of the genes (**A**) BCL6, (**B**) MYD88, (**C**) MYC, (**D**) IL7, (**E**) CCR4, and (**F**) CCR7 in the whole blood of prostate cancer patients undergoing IMRT (pink) and CK (red) radiotherapy treatment taken preexposure, 24 h after the 1st fraction, 5th/6th fraction (~1 week), and last fraction (4–8 weeks) and 1 month after the end of treatment. The box plot shows expression analysed in 11 IMRT and 11 CK prostate cancer patients. Expression levels are normalised to HPRT. Statistical analysis was performed on log-transformed data. * Significantly different (*p* < 0.05).

**Figure 5 ijms-25-01080-f005:**
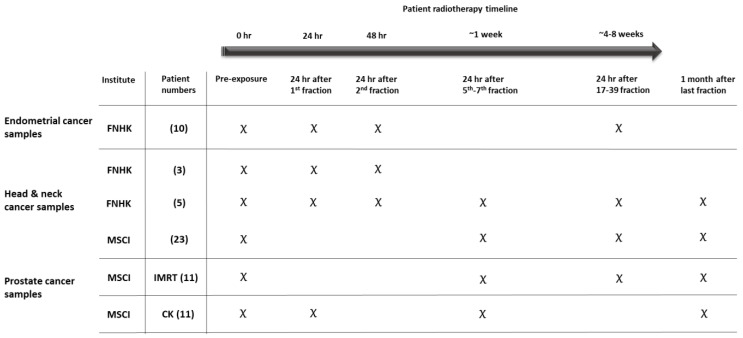
Patient radiotherapy timeline and blood collection. The section in grey illustrates the CyberKnife prostate cancer patients not included in bioinformatics analysis. MCSI refers to the Maria Sklodowska-Curie Institute and FNHK refers to Fakultní Nemocnice Hradec Králové.

**Table 1 ijms-25-01080-t001:** Combined analysis on nCounter data from endometrial, head and neck, and prostate patients across 2 timepoints (24 h and 5 weeks after 1st fraction) for all cancers identifying 15 significantly up- and 16 down-regulated genes of interest.

Common across All Cancers	Up-Regulated		Down-Regulated	
	15		16	
Genes	*ALOX5*	*NLRP3*	*CCR4*	*LTA*
	*BCL6*	*NOD2*	*CCR7*	*LTB*
	*CEBPB*	*TLR1*	*CD40*	*MAPKA*
	*CFD*	*TLR4*	*CD40LG*	*MAPKAPK5*
	*CXCL5*	*TLR5*	*ELK1*	*MYC*
	*LIMK1*	*TLR8*	*HMGN1*	*PRKCA*
	*MAPK14*	*TYROBP*	*IL23A*	*TCF4*
	*MYD88*		*IL7*	*TRADD*
			*IL8*	

**Table 2 ijms-25-01080-t002:** Patient radiotherapy dose and fractions.

Cancer Type	No. of Patients	Total Dose (Gy)	Dose per Fraction (Gy)	Number of Fractions
Endometrium	10	45	1.8	25
Head and neck	8 *	50–70	2–2.1	25–33
Head and neck	23	51–64.8	1.8–3	17–36
Prostate IMRT	11	78	2	39
Prostate CK	11	36.25	7.25	5

* 3 patients with 3 time point measurements.

## Data Availability

Research data are stored in an institutional repository and will be shared upon request to the corresponding author.

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
