# Peer review of "Transcriptional Inflammatory Signature in Healthy Donors and Different Radiotherapy Cancer Patients"

_ijms, 2024, doi:10.3390/ijms25021080_

Round 1

Reviewer 1 Report

Comments and Suggestions for Authors

In this article, the authors present the expression of inflammatory genes in normal and cancer patients after radiotherapy.

The authors present the aim of the study, but it is not fully clear to me. In the title, they mention biomarkers, but in the abstract, there is no information on biomarkers and the type of radiotherapy that was used.

In the introduction part, the authors reviewed other similar studies, but with a lower number of donors. Even though this study is larger, the differences in cancer patient numbers are high (10 vs. 31 vs. 22) and healthy donors. Once again if authors believe that this study is more reliable, they should explain these differences as well as limitations. The authors showed some limitations in the discussion parts tracking the age problem between donors. This explanation gives more arguments that the design of this study was not planned properly.

Then in the results part, after stratification to different groups, these data are difficult to explain and low reliable (see Table 1). I can not find the basic and new findings in the results part, compared to earlier articles.

In the Discussion part, even if the authors present findings, they are about inflammatory signatures/flu jabs which is inconsistent with the article title. Then, discuss particular genes rather than the findings.

Furthermore, I was interested in the introduction part, mostly these sentences:  "Therefore, the identification and monitoring of inflammatory genes in patients suffering from different malignancies undergoing RT might yield a prediction signature to anticipate radiotoxicity events and, subsequently, optimise and personalise radiotherapy  regimens and other treatment approaches in cancer patients". The authors did not elaborate on that in the next parts of the study and did not comment on how inflammatory blood biomarkers in blood might be connected with normal/cancer patients' responses after radiotherapy.

Taking all these arguments, at this stage I suggest rejecting or major revision of all parts of the article which will be consistent with the abstract and title.

Author Response

In this article, the authors present the expression of inflammatory genes in normal and cancer patients after radiotherapy.

We would like to thank the reviewer for taking the time to review our manuscript and providing very valuable feedback.

The authors present the aim of the study, but it is not fully clear to me. In the title, they mention biomarkers, but in the abstract, there is no information on biomarkers and the type of radiotherapy that was used.

We have taken in account the reviewer’s comments and have rewritten the abstract and title to make the aim of the study clearer and to avoid any ambiguities. Specifically, we have removed the word ‘Biomarkers from the title and abstract.

In the introduction part, the authors reviewed other similar studies, but with a lower number of donors. Even though this study is larger, the differences in cancer patient numbers are high (10 vs. 31 vs. 22) and healthy donors. Once again if authors believe that this study is more reliable, they should explain these differences as well as limitations. The authors showed some limitations in the discussion parts tracking the age problem between donors. This explanation gives more arguments that the design of this study was not planned properly.

Taking in account the reviewer comment, we have modified the introduction including a description of the differences between this study and previous smaller studies as well as indicating the limitations.

It wasn’t possible to match the average age of the healthy donors with the one of the cancer patients because it is difficult to find elderly donors available on the list of volunteers we have access to. Nevertheless, we have a similar male and female ratios.

Then in the results part, after stratification to different groups, these data are difficult to explain and low reliable (see Table 1). I cannot find the basic and new findings in the results part, compared to earlier articles.

In the results we described 3 new findings: First. (1) A radiation induced gene signature of 31 inflammation-associated genes was identified using a bigger cohort, second (2) Two confounding effects (infection by LPS and flu vaccine) were assessed to identify potential interaction which would have modified the radiation gene signature; the results clearly demonstrated that our signature is specific to radiation. (3) Besides, importantly, we observed in our healthy donors, that each donor has an individual inflammatory signature specific to each donor. This is a new finding.

In the Discussion part, even if the authors present findings, they are about inflammatory signatures/flu jabs which is inconsistent with the article title. Then, discuss particular genes rather than the findings.

We have now modified the title to take in account this valid point from the reviewer. The LPS and flu vaccine experiments were design to demonstrate the specificity of the gene signature identified to ionizing radiation. The aim of the manuscript is the radiation signature identified using the inflammatory gene panel in radiotherapy patient samples. So, we believe that the title shouldn’t include the parallel experiments performed to prove specificity. The new title should have now clarified the aim of the study; moreover, we have reorganized and modified the discussion to make clear that the main finding in this manuscript is the radiation induced gene signature.

We have also included some information from previous studies where some of the genes where already identified.

Furthermore, I was interested in the introduction part, mostly these sentences:  "Therefore, the identification and monitoring of inflammatory genes in patients suffering from different malignancies undergoing RT might yield a prediction signature to anticipate radiotoxicity events and, subsequently, optimise and personalise radiotherapy  regimens and other treatment approaches in cancer patients". The authors did not elaborate on that in the next parts of the study and did not comment on how inflammatory blood biomarkers in blood might be connected with normal/cancer patients' responses after radiotherapy.

We agree that this statement was unnecessary, and we have removed this sentence from the introduction as we understand it could mislead the reader. As mentioned above, we have also modified the introduction to clarify the aim and results of the project.

Taking all these arguments, at this stage I suggest rejecting or major revision of all parts of the article which will be consistent with the abstract and title.

Reviewer 2 Report

Comments and Suggestions for Authors

The authors present a valuable work, which demonstrates that monitoring the expression of specific inflammation-related genes in blood during the course of radiotherapy can provide relevant information which could be fed into a more personalised approach to radiotherapy. However, some details need to be focused as follows.

1.    In Figure 1, the composition of 23 healthy donors is inexplicable and need to be specified. In Figure 2, the pictures are not clear enough. In Figure 3, the meaning of asterisks does not be shown. It maybe *Significantly different (p < 0.05).

2.    For methodology, the methods of nCounter and MQRT-PCR are both targeting mRNA level, and the determination of protein level would confirm further the results of specific gene expression.

3.    The study may add the cancer controls receiving no treatment or surgical/chemical therapy to subdivide the patterns of different treatment methods.

Author Response

The authors present a valuable work, which demonstrates that monitoring the expression of specific inflammation-related genes in blood during the course of radiotherapy can provide relevant information which could be fed into a more personalised approach to radiotherapy. However, some details need to be focused as follows.

We would like to thank the reviewer for taking the time to review our manuscript and providing very valuable feedback.

  1. In Figure 1, the composition of 23 healthy donors is inexplicable and need to be specified. In Figure 2, the pictures are not clear enough. In Figure 3, the meaning of asterisks does not be shown. It maybe *Significantly different (p < 0.05).

Thank you for picking this up; In figure 1, we have modified the figure to indicate 23 healthy donors.

In figure 2, we have increased the size of the gene names for clarity, and we have included the list of genes on the results section.

The figure 3, the figure capture has been included together with the description of the asterisks. The colour of MYC's graph has also been corrected. 

  1. For methodology, the methods of nCounter and MQRT-PCR are both targeting mRNA level, and the determination of protein level would confirm further the results of specific gene expression.

The reviewer is correct, our analyses are providing transcriptomic information through different techniques, focusing on a particular set of genes involved in inflammatory pathways. We think that a continuation of the study looking at the protein level would be very interesting to identify biomarkers at the protein level, but we would like to point out that gene expression doesn’t always correlate with protein levels. There are numerous studies where very radiation responsive biomarkers of gene expression (O’Brien et al, 2018) are not very reliable at protein level (Nemzow, et al., 2023) and vice versa. 

O’Brien, G., Cruz-Garcia, L., Majewski, M. et al. FDXR is a biomarker of radiation exposure in vivo. Sci Rep 8, 684 (2018). https://doi.org/10.1038/s41598-017-19043-w

Nemzow L, Boehringer T, Bacon B, Turner HC. Development of a human peripheral blood ex vivo model for rapid protein biomarker detection and applications to radiation biodosimetry. PLoS One. 2023 Aug 10;18(8):e0289634. doi: 10.1371/journal.pone.0289634. PMID: 37561730; PMCID: PMC10414586.

  1. The study may add the cancer controls receiving no treatment or surgical/chemical therapy to subdivide the patterns of different treatment methods.

We included healthy controls in the study, and we also compared the pre-treatment samples from the cancer patients, and we could not find any differences between both sets of controls. None of the cancer patients included in the study received any chemotherapy or radiotherapy treatments before the start of the blood sampling. After the collection of the pre-treatment sample, cancer patients started their radiotherapy treatment.

Round 2

Reviewer 1 Report

Comments and Suggestions for Authors

The authors significantly changed the title, abstract, and the entire manuscript.  All my suggestions were considered and included in the revised version. I suggest only considering a small title change. In my opinion, the actual version is too general for readers. Maybe it is worth adding information like ...different cancer radiotherapy patients or ...endometrial, head and neck, and prostate cancer radiotherapy patients.